# Actions of FTY720 (Fingolimod), a Sphingosine-1-Phosphate Receptor Modulator, on Delayed-Rectifier K^+^ Current and Intermediate-Conductance Ca^2+^-Activated K^+^ Channel in Jurkat T-Lymphocytes

**DOI:** 10.3390/molecules25194525

**Published:** 2020-10-02

**Authors:** Wei-Ting Chang, Ping-Yen Liu, Sheng-Nan Wu

**Affiliations:** 1Institute of Clinical Medicine, College of Medicine, National Cheng Kung University, Tainan 70101, Taiwan; cmcvecho2@gmail.com (W.-T.C.); larry@mail.ncku.edu.tw (P.-Y.L.); 2Division of Cardiovascular Medicine, Chi-Mei Medical Center, Tainan 71004, Taiwan; 3Department of Biotechnology, Southern Taiwan University of Science and Technology, Tainan 71004, Taiwan; 4Division of Cardiology, Department of Internal Medicine, National Cheng Kung University Hospital, College of Medicine, National Cheng Kung University, Tainan 70401, Taiwan; 5Department of Physiology, National Cheng Kung University Medical College, Tainan 70101, Taiwan; 6Institute of Basic Medical Sciences, National Cheng Kung University Medical College, Tainan 70101, Taiwan; 7Department of Medical Research, China Medical University Hospital, China Medical University, Taichung 40402, Taiwan

**Keywords:** FTY720, lymphocytes, delayed-rectifier K^+^ current, inactivation kinetics, intermediate-conductance Ca^2+^-activated K^+^ channel

## Abstract

FTY720 (fingolimod), a modulator of sphingosine-1-phosphate receptors, is known to produce the immunomodulatory actions and to be beneficial for treating the relapsing multiple sclerosis. However, whether it exerts any effects on membrane ion currents in immune cells remains largely unknown. Herein, the effects of FTY720 on ionic currents in Jurkat T-lymphocytes were investigated. Cell exposure to FTY720 suppressed the amplitude of delayed-rectifier K^+^ current (*I*_K(DR)_) in a time- and concentration-dependent manner with an IC_50_ value of 1.51 μM. Increasing the FTY720 concentration not only decreased the *I*_K(DR)_ amplitude but also accelerated the inactivation time course of the current. By using the minimal reaction scheme, the effect of FTY720 on *I*_K(DR)_ inactivation was estimated with a dissociation constant of 3.14 μM. FTY720 also shifted the inactivation curve of *I*_K(DR)_ to a hyperpolarized potential with no change in the slope factor, and recovery from *I*_K(DR)_ became slow during the exposure to this compound. Cumulative inactivation for *I*_K(DR)_ in response to repetitive depolarizations was enhanced in the presence of FTY720. In SEW2871-treated cells, FTY720-induced inhibition of *I*_K(DR)_ was attenuated. This compound also exerted a stimulatory action on the activity of intermediate-conductance Ca^2+^-activated K^+^ channels in Jurkat T-lymphocytes. However, in NSC-34 neuronal cells, FTY720 did not modify the inactivation kinetics of KV3.1-encoded *I*_K(DR)_, although it suppressed *I*_K(DR)_ amplitude in these cells. Collectively, the perturbations by FTY720 on different types of K^+^ channels may contribute to the functional activities of immune cells, if similar findings appear in vivo.

## 1. Introduction

FTY720 (Fingolimod, Glienya^®^, 2-Amino-2-(2-[4-octylphenyl]ethyl)-1,3-propanediol) is a potent immunosuppressive agent that was originally derived from myriocin, a sphingosine-like fungal metabolite [1,2]. This compound is thought to possess encouraging efficacy profiles in different experimental autoimmune disorders and for the treatment of multiple sclerosis [1,3,4]. From another perspective, FTY720 can act to work through its binding to S1P receptor subtype 1 (S1P_1_), which can prevent the egress of lymphocytes from lymph nodes [5,6]. However, it is unclear whether this compound exerts its effect as a regulator of ion channels [7,8]. The K_V_1.3-encoded currents, which exhibit unique gating properties and voltage dependency, largely constitute the delayed-rectifier K^+^ currents (*I*_K(DR_) enriched in immune cells [9,10,11]. These currents, which can be functionally expressed in the plasma membrane and the inner mitochondrial membrane, have been demonstrated to perform key functions in the immune system such as lymphocyte activation [10,11,12,13,14,15]. In particular, the K_V_1.3 phenotype has been previously demonstrated to be specifically expressed in ion myelin-specific activated cells from patients with multiple sclerosis and not detected in T cells activated with other antigens [16]. K_V_1.3^+^ cells secrete considerable amounts of interferon-γ and tumor necrotic factor-α [17] and K_V_1.3 inhibitors potently suppress experimental autoimmune encephalitis [18,19]. The inhibitors of K_V_1.3 channels indeed represent a potentially therapeutic approach to the diseases including multiple sclerosis [2,4,15]. Additionally, it was noticed that the activity of intermediated-conductance Ca^2+^-activated K^+^ channels (i.e., KCa3.1-encoded channels) was functionally expressed in immune cells [20,21].

The Jurkat T cell line is a CD45-deficient clone derived from the E6-1 clone of the Jurkat human T-cell leukemic cell line, and it has been previously demonstrated to express K_V_1.3-type *I*_K(DR)_ [9,11,20]. This cell line is recognized as a suitable model for investigations of lymphocytic functions such as cellular activation and apoptosis [12]. The macroscopic *I*_K(DR)_ in Jurkat T lymphocytes is encoded by *KCNA3* gene giving rise to a native K_V_1.3 channel [9]. In these cells, recent studies have also demonstrated the presence of K_V_1.3 channels in the inner mitochondrial membrane, which is strongly linked to lymphocyte apoptosis because of the Bax-K_V_1.3 interaction [12,21,22,23].

Therefore, the goal of this study was to evaluate the effect of FTY720 on ionic current in Jurkat T-lymphocytes, particularly at *I*_K(DR)_ and IK_Ca_ channels. Unexpectedly, in these cells, we found that FTY720 not only diminished the peak amplitude of *I*_K(DR)_ (K_V_1.3-encoded current) but also shortened the time course of current inactivation in response to rapid membrane depolarization. Additionally, this agent was also effective at increasing the activity of IK_Ca_ channels in Jurkat T-lymphocytes. Therefore, it is anticipated that those actions caused by FTY720 may synergistically act to influence the recognition processes and subsequently to perturb the functional activities of immune cells and the immunological synapses in cell culture or in vivo [20,24].

## 2. Results

### 2.1. Inhibitory Effect of FTY720 on Delayed-Rectifier K^+^ Current (I_K(DR)_) Measured from Jurkat T-Lymphocytes

In this study, we initially evaluated the effect of FTY720 on *I*_K(DR)_ in Jurkat T-lymphocytes under whole-cell current recordings. As Jurkat cells were bathed in Ca^2+^-free Tyrode’s solution, the *I*_K(DR)_ in response to a 1-sec ramp voltage-clamp pulse from −90 to +90 mV could be readily elicited. This population of K^+^ currents has been previously identified as *I*_K(DR)_ and to resemble the K_V_1.3-encoded currents [9,10,11]. After cells were exposed to FTY720 at different concentrations, the amplitude of *I*_K(DR)_ evoked by the 1-sec long ramp pulse became progressively decreased (Figure 1). For example, at the level of +60 mV, FTY720 at a concentration of 3 μM significantly reduced *I*_K(DR)_ amplitude by 61 ± 3 % from 298 ± 13 to 122 ± 8 pA (n = 11, *P* < 0.05). After the washout of FTY720, the current amplitude returned to 282 ± 11 pA (n = 7). Additionally, nonactin (10 μM), a K^+^-selective neutral ionophore [25], could reverse FTY720-mediated inhibition of *I*_K(DR)_ seen in Jurkat T-lymphocytes.

The relationship between the FTY720 concentration and the relative amplitude of *I*_K(DR)_ was determined and then constructed. In these experiments, the examined cell was held at −50 mV, the ramp pulse from −90 to +90 mV with a duration of 1 sec was applied, and the *I*_K(DR)_ amplitudes were measured at the level of +60 mV in the presence of different FTY720 concentrations. As illustrated in Figure 1, FTY720 suppressed *I*_K(DR)_ amplitude in a concentration-dependent manner. By virtue of a non-linear least-squares fit to the data [26,27,28], the IC_50_ value required for the inhibitory effect of FTY720 on *I*_K(DR)_ in Jurkat T-lymphocytes was calculated to be 1.51 μM. These results reflect that FTY is capable of exerting a depressant action on *I*_K(DR)_ in these cells.

### 2.2. Kinetic Studies of FTY720-Induced Block of I_K(DR)_

Because the *I*_K(DR)_ inactivation in response to rapid and long-lasting depolarization tends to be enhanced (Figure 2), it is pertinent to evaluate the kinetics of FTY720-induced block seen in Jurkat T-lymphocytes. As illustrated in Figure 2, when cells were rapidly depolarized from −50 to +50 mV with a duration of 1 sec, it can be noted that exposure to FTY720 produced a reduction in the τ_inact_ value in a concentration-dependent fashion. Therefore, findings from these results strongly suggest that inhibitory effects of FTY720 on *I*_K(DR)_ in Jurkat T-lymphocytes can be explained in a satisfactory way by the state-dependent blocking where it can bind to the open state of the channel according to the first-order blocking scheme:

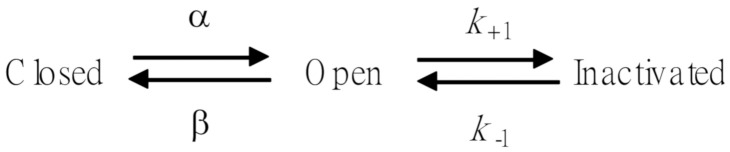

where α and β are kinetic constants for the opening and closing of K_V_ channel, and *k*_+1_ and *k*_−1_ are those for forward and reverse rate constants of the FTY720 binding. “Closed”, “Open”, and “Inactivated” represent the closed, open, and inactivated states of the channel, respectively. Forward and reverse rate constants, *k*_+1_ and *k*_−1_, were estimated from the values of τ_inact_ for FTY720-induced modification in the trajectory of *I*_K(DR)_ inactivation (Figure 2). In agreement with the kinetic scheme described under Materials and Methods, the relationship between *k*_+1_ and [B] was noted to become linear with a correlation coefficient of 0.96 (Figure 2C). The forward and reverse rate constants obtained from 8 to 10 different cells were calculated to be 1.0046 sec^−1^μM^−1^ and 3.1558 sec^−1^, respectively. On the basis of these rate constants, a value of 3.14 μM for the dissociation constant (*K*_D_ = *k*_−1_/*k*_+1_) of this compound was derived. This value was found to be in agreement with the IC_50_ value determined by the concentration-response curve (Figure 1B).

### 2.3. Steady-State Inactivation of I_K(DR)_ During the Exposure to FTY720

To further characterize the inhibitory effect of FTY720 on *I*_K(DR)_, we next investigated the voltage dependence of the effect of FTY720 on *I*_K(DR)_ in Jurkat T-lymphocytes by using a two-step voltage protocol. Figure 3 illustrates the steady-state inactivation curve of *I*_K(DR)_ with or without the addition of 3 μM FTY720. In these experiments, a 1-sec conditioning pulse to different potentials preceded the test pulse to +50 mV from a holding potential of −50 mV. The relationship between the conditioning potentials and the normalized amplitudes of peak *I*_K(DR)_ were constructed and fitted with a Boltzmann function, as described in Materials and Methods. In the control (i.e., in the absence of FTY720), the voltages for half-maximal inactivation (*V*_1/2_) and corresponding slope factor (*k*) were −5.4 ± 0.2 mV and 5.31 ± 0.02 mV (*n* = 9), respectively, while in the presence of 3 μM FTY720, the values of *V*_1/2_ and *k* were -18.5 ± 0.9 mV and 5.32 ± 0.02 mV (*n* = 8), respectively. It is clear from these results that the addition of FTY720 (3 μM) significantly shifted the midpoint of the inactivation curve toward hyperpolarizing voltage by approximately 13 mV (*P* < 0.05); however, no discernible change in the slope factor (i.e., *k*) was observed in the presence of this compound. Therefore, in Jurkat T-lymphocytes, there was a voltage-dependence of the steady-state inactivation curve of *I*_K(DR)_ during cell exposure to FTY720.

### 2.4. Effect of FTY720 on the Recovery of I_K(DR)_ from Inactivation

The effect of FTY720 on the recovery of *I*_K(DR)_ from inactivation was further determined with another two-pulse protocol consisting of a first (conditioning) depolarizing pulse, which was enough to enable the block to reach a steady state. In this set of experiments, the membrane potential was initially taken to +50 mV from −50 mV for a duration of 150 msec, after which a second depolarizing pulse (test pulse) was then applied at the same potential as the conditioning pulse (Figure 4). The ratios of the peak amplitude of *I*_K(DR)_ evoked in the response to the test and the conditioning pulses were then determined as a measure of recovery from block, and plotted against the interpulse interval. As depicted in Figure 4, the time course of recovery from inactivation in the control was fitted to a single exponential function with a time constant of 1.8 ± 0.1 sec (*n* = 9). As FTY720 at a concentration of 1 and 3 μM was applied, the time constant was significantly prolonged to 3.1 ± 0.2 (*n* = 7, *P* < 0.05) and 4.8 ± 0.2 sec (*n* = 7, *P* < 0.05). Therefore, the recovery of *I*_K(DR)_ from inactivation observed in Jurkat T-lymphocytes clearly became slowed in the presence of FTY720. The delayed recovery of *I*_K(DR)_ caused by FTY720 was most likely due to the open channel block.

### 2.5. FTY720-Induced Increase in Cumulative Inhibition of I_K(DR)_ Inactivation

In another set of experiments, we sought to determine whether FTY720 can alter the time course of *I*_K(DR)_ inactivation in response to repetitive stimuli in Jurkat T-lymphocytes. Under control condition, a single 1-sec depolarizing step to +50 mV from a holding potential produced an exponential decline with a time constant of 352 ± 12 msec (*n* = 11). However, the time constant for 1-sec repetitive pulses to +50 mV, each of which lasted 40 msec with 10-msec interval at −50 mV between the depolarizing pulses, was significantly reduced to 326 ± 11 msec (*n* = 10). Moreover, as depicted in Figure 5, there was a progressive increase in the decline of *I*_K(DR)_ in response to rapid depolarizing stimuli. As the cells were exposed to FTY720, the value of time constant taken from such train of short repetitive pulses was further decreased. The addition of 1 and 3 μM FTY720 significantly decreased the time constant to 212 ± 8 msec (n = 8, *P* < 0.05) and 156 ± 8 msec (*n* = 8, *P* < 0.05), respectively. Therefore, findings from these results indicate that an excessive accumulative inactivation of *I*_K(DR)_ during cell exposure to FTY720 can be detected.

### 2.6. Effect of FTY720 on I_K(DR)_ in Jurkat T-Lymphocytes Treated with SEW2781

FTY720 was recognized to be an agent involving modulation of S1P receptors on lymphocytes [5,6]. We next evaluate whether binding to the S1P receptor can influence the FTY720-induced block of *I*_K(DR)_ in Jurkat T-lymphocytes. In this set of experiments, Jurkat cells were preincubated with SEW2871 (3 μM) for 5 h. SEW2871 is a selective agonist of the S1P_1_ receptor [29]. In SEW2871-treated cells, the inhibitory effect of FTY720 on *I*_K(DR)_ was attenuated (Figure 6). Distinguishable from the results from control cells (i.e., SEW2871 was not pre-treated), the inactivation time course of *I*_K(DR)_ remained unaltered in the presence of FTY720. Similar actions occurred in Jurkat T-lymphocytes treated with S1P (10 μM). The magnitude of FTY720-induced inhibition of *I*_K(DR)_ was diminished with no modification in activation or inactivation kinetics of this current. These results led us to propose that the inhibitory effect of FTY720 on *I*_K(DR)_ in these cells tends to be associated with its binding to the S1P_1_ receptor.

### 2.7. Stimulatory Effect of FTY720 on the Activity of IK_Ca_ Channels in Jurkat T-Lymphocytes

In another set of experiments, we sought to determine whether FTY720 exerts any effect on the activity of IK_Ca_ channels in these cells. Cell-attached current recordings were performed and cells were bathed in high K^+^-solution containing 1.8 mM CaCl_2_. As shown in Figure 7, under our experimental conditions, the activity of IK_Ca_ channels, which was sensitive to block by TRAM-34 and to stimulation by 9-phenanthrol, was readily detected as described previously in different types of cells [25,30,31]. As FTY720 was applied to the bath, the probability of IK_Ca_-channel openings was progressively raised. For example, the addition of FTY720 at a concentration of 1 and 3 μM significantly increased the channel open probability measured at -60 mV to 0.0035 ± 0.007 and 0.0047 ± 0.009, respectively, from control of 0.0013 ± 0.0005 (*n* = 11, *P* < 0.05). Conversely, no clear change in single-channel conductance of IK_Ca_ channels was found in the presence of FTY720 (48 ± 2 pS [control] versus 49 ± 2 pS [3 μM FTY720], *n* = 9, *P* > 0.05). In the continued presence of FTY720 (3 μM), subsequent addition of TRAM-34 (1 μM) reversed the probability of channel openings, as evidenced by the significant decrease of channel activity to 0.0018 ± 0.0005 (*n* = 9, *P* < 0.05). However, paxilline (1 μM), known to be a blocker of BK_Ca_ channels, did not produce any effect on FTY720-induced increase of IK_Ca_-channel activity. Therefore, apart from the inhibition of *I*_K(DR)_, FTY720 is effective at stimulating the activity of IK_Ca_ channels in Jurkat T-lymphocytes.

### 2.8. Effect of FTY720 on I_K(DR)_ Recorded from NSC-34 Neuronal Cells

In a final set of experiments, we further investigated whether this compound had any effect on *I*_K(DR)_ in other types of cells (e.g., NSC-34 neuronal cells). Cells were bathed in Ca^2+^-free Tyrode’s solution containing 1 μM tetrodotoxin and 0.5 mM CdCl_2_ and the recording pipette was filled with K^+^-containing solution. Tetrodotoxin and CdCl_2_ were used to block Na^+^ and Ca^2+^ currents, respectively. Under this condition, the macroscopic *I*_K(DR)_ in response to membrane depolarization has been previously described in these cells and identified primarily as K_V_3.1-encoded current [32]. Unexpectedly, when cells were exposed to FTY720 (3 μM), the amplitude of *I*_K(DR)_ was slightly decreased; however, neither activation nor inactivation time course of *I*_K(DR)_ in response to membrane depolarization was changed in its presence (Figure 8). These data are clearly distinguishable from those observed in Jurkat T-lymphocytes described above.

## 3. Discussion

In this study, we provide the evidence to show that in Jurkat T-lymphocytes, FTY720 produces an inhibitory effect on *I*_K(DR)_ in a concentration-, voltage- and state-dependent fashion. According to both the concentration-dependent relationship and minimal reaction scheme described herein, the values of IC_50_ and *K*_D_ required for FTY720-mediated inhibition of *I*_K(DR)_ were measured from Jurkat T-lymphocytes were calculated to be 1.51 and 3.14 μM, respectively. The steady-state inactivation curve of *I*_K(DR)_ in the presence of FTY720 (3 μM) produced a leftward shift by approximately 13 mV, indicating that such inhibitory action is voltage-dependent. There might thus be a pertinent link between its effects on lymphocytes and the inhibitory effect on *I*_K(DR)_ (i.e., K_V_1.3-encoded current). However, whether the ability of this compound to suppress mitochondrial K_V_1.3-encoded channels and to perturb the Bax-K_V_1.3 interaction [21,22,23] has yet to be clearly demonstrated.

The rate of *I*_K(DR)_ inactivation was considerably enhanced as the FTY720 concentration increased. In our study, recovery from inactivation in the presence of FTY720 apparently became slow, because more than 10 sec was required for the channel to recover completely. It is thus possible that K_V_1.3 channel, once open, can be subsequently plugged by the FTY720 molecule in solution. Assuming that periodical changes in membrane potential of inner mitochondrial membranes may occur [33], the issue of whether blockade of mitochondrial K_V_1.3 channels caused by FTY720 is responsible for its effects on Bax-K_V_1.3 interactions [21,22,23] is worthy of being investigated. FTY720 has indeed been reported to mediate cytochrome c release from mitochondria in rat thymocytes [34]. Nonetheless, because of the importance of *I*_K(DR)_ (i.e., K_V_1.3-encoded current) in contributing to functional activities of lymphocytes such as apoptosis [12,34], the effects of FTY720 presented herein could provide novel insights into pharmacological or immunological properties of FTY720 and other structurally related compounds [15,35].

After daily administration of FTY720 over 1 week (5 mg/day), the blood concentration of FTY720 could be around 18.2 ng/mL (0.053 μM) [36]. Together with a quantitative description of the inactivation time course of *I*_K(DR)_ in response to rapid depolarization during cell exposure to different FTY720 concentrations, our results provide the evidence to indicate the FTY720 molecule can act as a state-dependent open-channel blocker of K_V_ channels enriched in Jurkat T-lymphocytes. Any changes of *I*_K(DR)_ amplitude or kinetics caused by FTY720 depend not only on the FTY720 concentration given, but also on membrane potential and intracellular Ca^2+^ concentrations. The inhibitory effect of FTY720 on the amplitude and gating of *I*_K(DR)_ in lymphocytes may thus occur at a concentration achievable in humans. However, this compound exerted a slight inhibition of K_V_3.1-encoded *I*_K(DR)_ and did not modify the inactivation kinetics of this current in NSC-34 neuronal cells.

FTY720 is recognized as a modulator of S1P receptors. In this study, in Jurkat T-lymphocytes preincubated with either SEW2871 or S1P, the magnitude of FTY-induced block of *I*_K(DR)_ was diminished with no change in activation or inactivation kinetics of this current. SEW2871 was previously reported to be an agonist of the S1P_1_ receptor [29]. It is thus tempting to speculate that FTY720-mediated changes in the amplitude and gating of *I*_K(DR)_ are strongly linked to its binding to S1P receptors inherently in Jurkat cells. In Coppi and colleagues’ work, through inhibiting potassium currents and positively modulating S1P synthesis, A_2B_ receptor (A_2B_R) suppressed cell differentiation in cultured oligodendrocytes progenitor cells [37]. S1P inhibits delayed rectifier potassium current of guinea pigs’ ventricular myocytes. Nevertheless, there is no previous literature reporting the effects of S1P on IK(DR) of immune cells which may potentially contribute to the immune-modulation. Alternatively, whether the presence of FTY720 can perturb ionic currents in lymphocytes derived from K_V_1.3 null mice remains to be interestingly explored.

Another noteworthy finding in this study is that in Jurkat T-lymphocytes, the activity of IK_Ca_ (or KCNN4-encoded) channels was functionally expressed and subject to stimulation by 9-phenanthrol and to inhibition by TRAM-34 [25,30,31,38]. Additionally, the exposure to FTY720 was effective at enhancing the activity of IK_Ca_ channels with no change in the single-channel conductance of these channels. TRAM-34, still in the continued presence of FTY720, suppressed FTY720-mediated increase of IK_Ca_-channel activity; however, paxilline did not exert any effect on FTY720-induced stimulation of IK_Ca_ channels. Consistent with previous observations in neutrophils [39,40], we found that the activity of BK_Ca_ channels was absent in Jurkat T-lymphocytes. However, to what extent FTY720-induced activation of IK_Ca_ channels found in this study is capable of exerting its effects on the activity of T lymphocytes remains to be further investigated.

Additionally, in this study, the K_V_3.1-encoded current, most of which constitute *I*_K(DR)_ in NSC-34 neuronal cells, was relatively insensitive to blocking by FTY720. Neither the activation nor inactivation time course of *I*_K(DR)_ in NSC-34 cells was altered during the exposure to FTY720. FTY720-induced effects on the amplitude and gating of K_V_1.3-encoded *I*_K(DR)_ observed in Jurkat T-lymphocytes thus tends to be selective and can have therapeutic relevance.

In human T-lymphocytes, cell exposure to FTY720 also suppressed *I*_K(DR)_ amplitude as well as shortened the inactivation time course of the current. The FTY720-induced inhibition of *I*_K(DR)_ was attenuated post the treatment of SEW2871. It speculated that FTY720-mediated changes in the amplitude and gating of *I*_K(DR)_ are associated with its binding to S1P receptors inherently. The graphic abstract summarized the major findings that through perturbing different types of K^+^ channels FTY720 may contribute to the functions of immune cells and result in an immune-modulating effect. Taken together, despite the detailed mechanism of FTY720 actions on these K_V_ channels being unclear, if similar findings are experimentally made in lymphocytes occurring in cell culture or in vivo to those presented herein, the additional actions of FTY720 or its structurally related agents described here will result in significant additional changes in immune reactions [3,5,15]. Since this compound is regarded as an immune modulator [6,7], the FTY720-perturbed effects on ionic currents demonstrated herein could potentially participate in various clinical disorders, such as multiple sclerosis.

## 4. Materials and Methods

### 4.1. Drugs and Solutions

FTY720 (Fingolimod, Gilenya^®^, 2-amino-2-(2-[4-octylphenyl]ethyl)-1,3-propanediol hydrochloride, C_19_H_33_NO_2_), phytohemagglutinin (PHA), sphingosine-1-phosphate (S1P), tetrodotoxin and trypan blue were obtained from Sigma-Aldrich (St. Louis, MO, USA), nonactin, 9-phenanthrol, SEW2871 (5-[4-phenyl-5-(trifluoromethyl)thiophen-2-yl]-3-[3-(trifluoromethyl)phenyl]1,2,4-oxadiazole) and TRAM-34 (1-((2-chlorophenyl)(diphenyl))methyl)-1*H*-pyrazole) were from Tocris Cookson Ltd. (Bristol, UK), and paxilline was from Alomone Labs (Jerusalem, Israel). Chlorotoxin and margatoxin were kindly provided by Dr. Woei-Jer Chuang, (Department of Biochemistry, National Cheng Kung University Medical College, Tainan, Taiwan). Tissue culture media, fetal bovine serum (FBS), L-glutamine, trypsin/EDTA and penicillin-streptomycin were obtained from Invitrogen (Carlsbad, CA, USA). All other chemicals including CdCl_2_ were commercially available and of reagent grade. The twice-distilled water that had been de-ionized through a Milli-Q water purification system (APS Water Services Inc., Van Nuys, CA, USA) was used in all experiments.

The composition of normal Tyrode’s solution was 136.5 mM NaCl, 5.4 mM KCl, 1.8 mM CaCl_2_, 0.53 mM MgCl_2_, 5.5 mM glucose, and 5.5 mM HEPES-NaOH buffer; pH 7.4. To measure whole-cell K^+^ currents or membrane potential, a patch pipette was filled with the following solution: 140 mM KCl, 1 mM MgCl_2_, 3 mM Na_2_ATP, 0.1 mM Na_2_GTP, 0.1 mM EGTA, and 5 mM HEPES-KOH buffer; pH 7.2. To avoid possible contamination of Cl^−^ currents, Cl^−^ ions inside the pipette solution was replaced with aspartate. For the recordings of BK_Ca_- or IK_Ca_-channel activity, a high K^+^-bathing solution contained 145 mM KCl, 0.53 mM MgCl_2_, and 5 mM HEPES-KOH buffer; pH 7.4, and the pipette solution contained 145 mM KCl, 2 mM MgCl_2_ and 5 mM HEPES-KOH buffer; pH 7.2.

### 4.2. Cell preparations

The Jurkat T cell line, a human T cell lymphoblast-like cell line (clone E6-1), was obtained from the Bioresource Collection and Research Center (BCRC-60255; Hsinchu, Taiwan). Cells were routinely grown in a RPMI-1640 medium supplemented with 10% heat-inactivated FBS (*v*/*v*), 100 U/mL penicillin and 10 μg/mL streptomycin. They were maintained at 37 °C in a 95% air and 5% CO_2_ humidified atmosphere. In a separate set of experiments, Jurkat T-lymphocytes were incubated with S1P (10 μM) or SEW2871 (3 μM) at 37 °C for 5 h. NSC-34 neuronal cells were produced by the fusion of motor neuron-enriched, embryonic mouse spinal cords with mouse neuroblastoma. They were maintained in a 1:1 mixture of DMEM and Ham’s F12 medium supplemented with 10% FBS and 1% penicillin-streptomycin. To slow cell proliferation and enhance their maturation towards a differentiated state, before confluence, NSC-34 cells were grown in 1:1 DMEM plus Ham’s F12 medium supplemented with 1% FBS for 48 h. The viability of these cells was often evaluated by the trypan blue-exclusion test. The experiments were made five or six days after cells had been cultured (60–80% confluence).

For preparations of human T-lymphocytes, CD3^+^ and CD4^+^ lymphocytes were isolated from healthy donors by E-rosetting (StemCell Technology, Vancouver, BC, Canada) and Ficoll-Plaque density-gradient centrifugation (ICN Biomedicals, Aurora, OH, USA). Freshly isolated human T-lymphocytes were preactivated with 4 μg/mL PHA for about four hours. Blood intended for isolation of T-lymphocytes was obtained from healthy volunteers.

### 4.3. Electrophysiological Measurements

Jurkat T-lymphocytes or NSC-34 neuronal cells were harvested with 1% trypsin/EDTA solution prior to the experiments and a small aliquot of cell suspension was transferred to a home-made recording chamber positioned on the stage of a DM-IL inverted microscope (Leica Microsystems, Wetzlar, Germany) coupled to digital video system (DCR-TRV30; Tokyo, Japan) with a magnification of up to 1500×. Cells were immersed at room temperature (20–25 °C) in normal Tyrode’s solution containing 1.8 mM CaCl_2_. The recording pipettes were pulled from Kimax-51 glass capillaries (#34500; Kimble Glass, Vineland, NJ) using either a PP-83 puller (Narishige, Tokyo, Japan) or a horizontal puller (Sutter Instruments model P-97; Novato, CA, USA), and their tips were then fire-polished with a microforge (MF-83; Narishige). As the pulled pipettes were filled with different internal solutions described above, their resistance commonly ranged between 3 and 5 MΩ. During each experiment, the pipette mounted maneuvered by using a WR-98 hydraulic micromanipulator (Narishige, Amityville, NY, USA). Current signals recorded under cell-attached, inside-out, or whole-cell configuration were measured with the standard patch-clamp technique by use of either an RK-400 amplifier (Bio-Logic, Claix, France) or an Axopatch-200B amplifier (Molecular Devices, Sunnyvale, CA, USA) [11,25]. Liquid junction potential existing between the internal pipette solution and the extracellular medium was corrected immediately before seal formation was made.

### 4.4. Data Recordings

The signals, consisting of potential and current traces, were stored online on a TravelMate-6253 laptop computer (Acer, Taipei, Taiwan) at 10 kHz through a Digidata-1550A interface (Molecular Devices). During the experiments, the latter device is controlled by pCLAMP 10.2 software (Molecular Devices, San Jose, CA, USA). Current signals were low-pass filtered at 3 kHz. Through digital-to-analogue conversion, the voltage-step profiles of rectangular or ramp pulses created from pCLAMP 10.2 were commonly used to determine the current–voltage (I–V) relations for different types of ionic currents (e.g., *I*_K(DR)_). Some signals digitally stored through either wired USB or wireless Bluetooth were further analyzed using different analytical tools which include OriginPro 2016 (OriginLab, Northampton, MA, USA), Labchart 7.0 program (AD Instruments; Gerin, Tainan, Taiwan) and custom-made macros built in an Excel 2013 spreadsheet under Windows 7 (Microsoft, Redmond, WA, USA). The details of data analyses were listed in Appendix A.

### 4.5. Analyses of Single IK_Ca_-channel Currents

Single BK_Ca_- or IK_Ca_-channel currents were analyzed using pCLAMP 10.2 software (Molecular Devices, San Jose, CA, USA). Single-channel amplitudes were determined by fitting Gaussian distributions to the amplitude histograms of the closed and open states. The probabilities of channel openings were broadly estimated using an iterative process to minimize the χ^2^ values calculated with a sufficiently large number of independent events. Channel activity was defined as N·P_O_, the product of the channel number (*N*) and open probability (*P*_O_). Single-channel conductance of IK_Ca_ channels with or without the addition of FTY720 was determined by linear regression using mean values of current amplitudes measured at different potentials. The lifetime distributions of open or closed states were fitted with logarithmically scaled bin width [25].

### 4.6. Statistical Analyses

By using the least-squares minimization procedure [26], linear or nonlinear curve-fitting to experimental data in this study was performed with pCLAMP 10.2 (Molecular Devices, San Jose, CA, USA), the “Solver” add-in bundled with Excel 2013 (Microsoft, Redmond, WA, USA), or OriginPro 2016 (OriginLab, Northampton, MA, USA). The macroscopic and single-channel current data are expressed as the mean ± SEM with sample sizes (n) indicating the number of cells from which the results were taken, and error bars are plotted as SEM. The paired or unpaired Student’s t-test and one-way analysis of variance of the least-significant-difference method for multiple comparisons were used to evaluate the statistical difference among means. Statistical analyses were performed using IBM SPSS version 20.0 (IBM Corp., Armonk, NY, USA). A *P* value of less than 0.05 was considered to indicate the statistical difference.

## Figures and Tables

**Figure 1 molecules-25-04525-f001:**
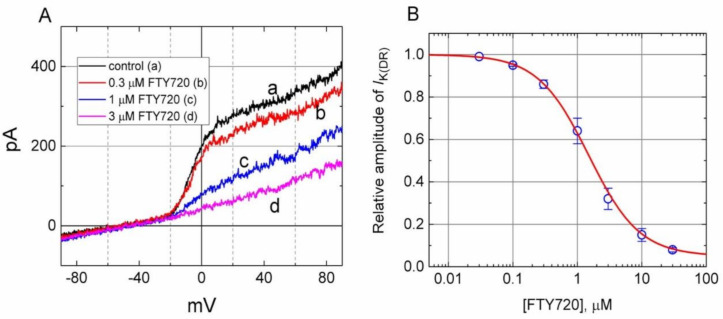
Inhibitory effect of FTY720 on delayed-rectifier K^+^ current (*I*_K(DR)_) in Jurkat T-lymphocytes. In these experiments, cells were immersed in Ca^2+^-free Tyrode’s solution, the recording pipette was filled with a K^+^-containing solution, and the ramp pulses from −90 to +90 mV with a duration of 1 sec were applied. (**A**) Superimposed current traces in response to ramp pulse obtained in the absence (a) and presence of 0.3 μM, 1 μM and 3 μM FTY720. (**B**) Concentration-response curve of FTY720-induced inhibition of *I*_K(DR)_ appearing in these cells. The smooth line represents the best fit to the modified Hill equation as described under Materials and Methods. The IC_50_ value, maximally inhibited percentage of *I*_K(DR)_, and Hill coefficient for FTY720-induced inhibition of *I*_K(DR)_ were 1.51 μM, 95% (i.e., *a* = 0.05) and 1.1, respectively. Each point represents the mean ± SEM (*n* = 9–13).

**Figure 2 molecules-25-04525-f002:**
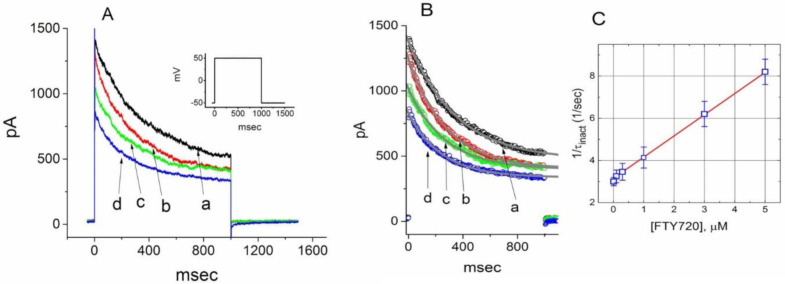
Evaluation of the inactivation kinetics of FTY720-induced block of *I*_K(DR)_ in Jurkat T-lymphocytes. In (**A**), the time courses of current inactivation in the absence (a) and presence of 0.3 μM (b), 1 μM (c) and 3 μM (d) FTY720 were well fitted by a single exponential as indicated in (**B**). Inset in (**A**) indicates the voltage protocol used. In (**C**), the kinetics of FTY-720-induced block of *I*_K(DR)_ in Jurkat T-lymphocytes was evaluated. The reciprocal of inactivation time constant of *I*_K(DR)_ (i.e., 1/τ_inact_) taken by exponential fits of the trajectory of *I*_K(DR)_ inactivation was constructed and plotted as a function of the FTY720 concentration (mean ± SEM; *n* = 8–10 for each point). Data points were well fitted by linear regression, indicating that FTY720-induced blocking tends to occur with a molecularity of 1. Blocking (*k*_+1_) and unblocking (*k*_−1_) rate constants, respectively derived from the slope and the *y*-axis intercept of the interpolated line, were estimated to be 1.0046 sec^−1^μM^−1^ and 3.1558 sec^−1^, respectively.

**Figure 3 molecules-25-04525-f003:**
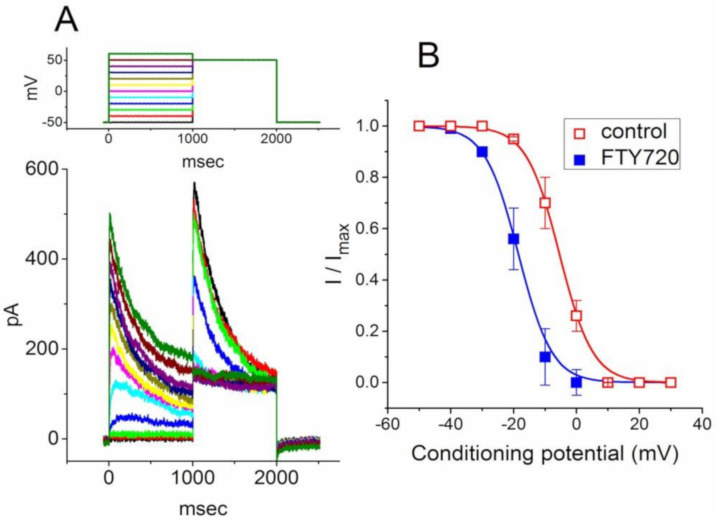
Effect of FTY720 on steady-state inactivation of *I*_K(DR)_ recorded from Jurkat T-lymphocytes. (**A**) Superimposed current traces obtained in the control. The conditioning voltage pulses with a duration of 1 sec to various membrane potentials ranging from −50 to +30 mV in 10-mV increments were applied from a holding potential of −50 mV. Following each conditioning pulse, a test pulse to +50 mV with a duration of 1 sec was applied to elicit *I*_K(DR)_. The upper part indicates the voltage protocol applied. (**B**) Steady-state inactivation curves of *I*_K(DR)_ taken in the absence (□) and presence (■) of 3 μM FTY720 (mean ± SEM; *n* = 7–10 for each point). Note that the presence of FTY720 shifts the midpoint of the inactivation curve toward hyperpolarizing voltage with no clear change in the slope factor.

**Figure 4 molecules-25-04525-f004:**
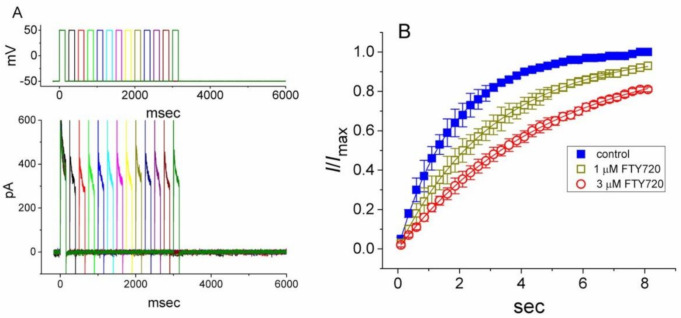
Effect of FTY720 on the recovery of *I*_K(DR)_ from inactivation in Jurkat T-lymphocytes. Changes in currents were measured under two-step voltage-clamp conditions. Cells were depolarized from −50 to +50 mV with a duration of 150 msec and various interpulse durations were applied. (**A**) Superimposed current traces obtained in the presence of 3 μM FTY720. The upper part indicates the voltage protocol used. (**B**) Time courses for the recovery of *I*_K(DR)_ from inactivation obtained in the absence (■) and presence of 1 μM (□) and 3 μM (○) FTY720. Each point represents the mean ± SEM (n = 6–8). Each smooth line is the best fit to the data.

**Figure 5 molecules-25-04525-f005:**
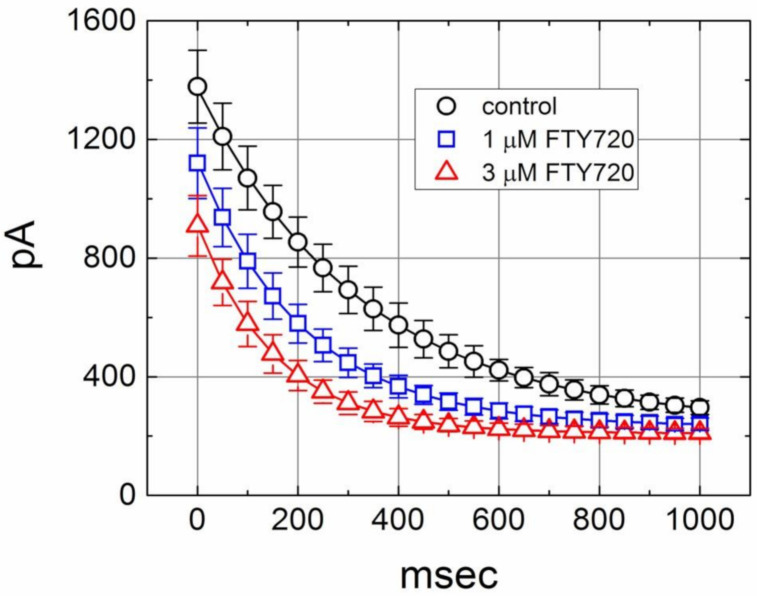
Excessive accumulative inactivation of *I*_K(DR)_ during repetitive stimuli in the absence (○) and presence of 1 μM (□) and 3 μM (△) FTY720 recorded from Jurkat T-lymphocytes. Currents were obtained during 1-sec repetitive depolarizations to +50 mV from a holding potential of −50 mV. Notably, the presence of FTY720 increases the rate of excessive accumulative inactivation of *I*_K(DR)_ in response to repetitive stimuli.

**Figure 6 molecules-25-04525-f006:**
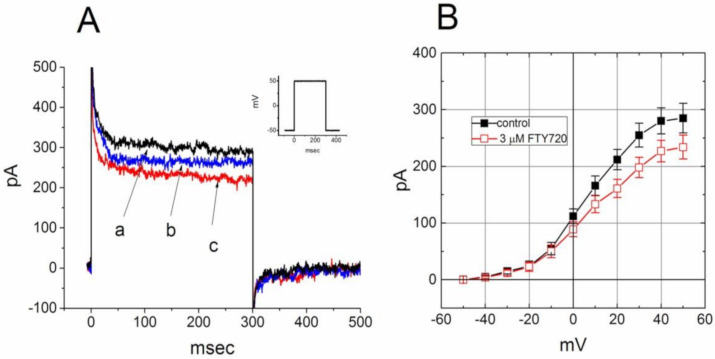
Effect of FTY720 on *I*_K(DR)_ in Jurkat T-lymphocytes treated with SEW2871. Cells were preincubated with SEW2871 (3 μM) for 5 h. (**A**) Superimposed *I*_K(DR)_ traces obtained in the (a) absence and presence of (b) 1 and (c) 3 μM FTY720. Inset indicates the voltage protocol used. (**B**) Averaged *I-V* relationships of *I*_K(DR)_ in the absence (■) and presence (□) of 3 μM FTY720 (mean ± SEM; *n* = 8–10 for each point). In each cell examined, *I*_K(DR)_ was elicited from −50 mV to different voltages ranging from −50 to +50 mV in 10-mV increments.

**Figure 7 molecules-25-04525-f007:**
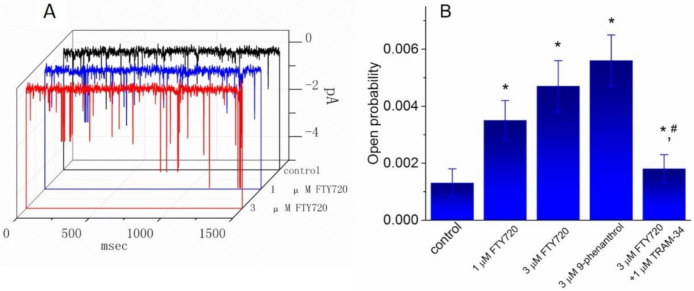
Effect of FTY720, 9-phenanthrol and FTY720 plus TRAM-34 on the activity of IK_Ca_ channels recorded from Jurkat T-lymphocytes. Cells were bathed in a high-K^+^ bathing solution containing 1.8 mM CaCl_2_. In these experiments, cell-attached current recordings were performed and the probabilities of channel openings at −60 mV with or without the addition of various agents were measured and compared. (**A**) Original single-channel current traces obtained in the control and during exposure to 1 μM or 3 μM FTY720. Downward deflection indicates the opening event of the IK_Ca_ channel. (**B**) Summary of the data showing the effect of FTY720, 9-phenanthrol, and FTY720 plus TRAM-34 on IK_Ca_-channel activity in Jurkat T-lymphocytes. Values are means ± SEM for *n* = 8–11 cells in each bar. * Significantly different from control (*P* < 0.05) and # significantly different from FTY720 (3 μM) alone group (*P* < 0.05).

**Figure 8 molecules-25-04525-f008:**
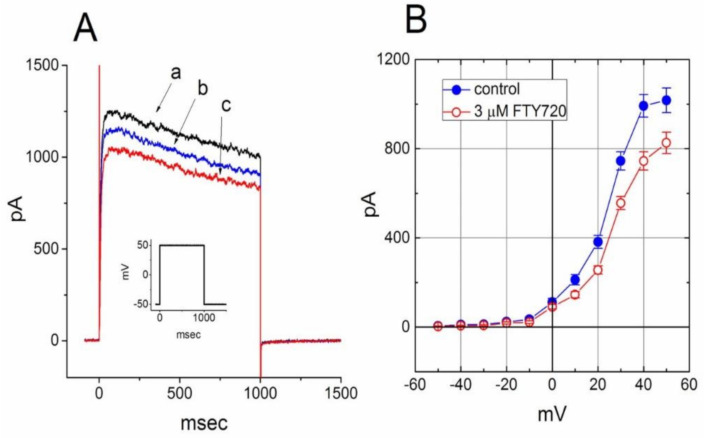
Effect of FTY720 on *I*_K(DR)_ in NSC-34 neuronal cells. Cells were bathed in Ca^2+^-free Tyrode’s solution containing 1 μM tetrodotoxin and 0.5 mM CdCl_2_ and the recording pipette was filled with K^+^-containing solution_._ (**A**) Superimposed *I*_K(DR)_ traces obtained in the absence (a) and presence of 1 μM (b) and 3 μM (c) FTY720. The inset indicates the voltage protocol used. (**B**) Averaged *I-V* relationship of *I*_K(DR)_ in the absence (●) and presence (○) of 3 μM FTY720 (mean ± SEM; *n* = 7–10 for each point). The *I*_K(DR)_ amplitude was measured at the end of each depolarizing pulse. Notably, cell exposure to FTY slightly deceased *I*_K(DR)_ amplitude with no modifications in activation or inactivation kinetics of the current.

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
