# Peer review of "Actions of FTY720 (Fingolimod), a Sphingosine-1-Phosphate Receptor Modulator, on Delayed-Rectifier K+ Current and Intermediate-Conductance Ca2+-Activated K+ Channel in Jurkat T-Lymphocytes"

_molecules, 2020, doi:10.3390/molecules25194525_

Round 1

Reviewer 1 Report

In this manuscript researchers gave a great biophysical analysis about the actions of FTY720 on various K+ channels in Jurkat T-Lymphocytes. The electrophysiological analysis is comprehensive, complete, well presented and well-performed. Sounds interesting and results could be important in translational research.

However, as minor comment, reviewer suggests to add a "translational" section to the discusson where researchers explain the clinical relevance of the study. That section would be improve the quality of the manuscript.

As major suggestion, reviewer would ask researchers to perform at least one in vitro functional assay supporting FTY720 dependent electrophisiological changes have impact on the effector functions of Jurkat T-lymphocytes. 

Author Response

Reviewer #1:

In this manuscript researchers gave a great biophysical analysis about the actions of FTY720 on various K+ channels in Jurkat T-Lymphocytes. The electrophysiological analysis is comprehensive, complete, well presented and well-performed. Sounds interesting and results could be important in translational research.

Question 1. However, as minor comment, reviewer suggests to add a "translational" section to the discussion where researchers explain the clinical relevance of the study. That section would be improve the quality of the manuscript.

Reply 1. As advised by the reviewer, another sentence was added to the end of the Discussion section of the revised manuscript. That is, “Since this compound is regarded as an immune modulator, the FTY720-perturbed effects on ionic currents demonstrated herein could potentially participate in various disorders, such as multiple sclerosis or cytokine storms.”

Question 2. As major suggestion, reviewer would ask researchers to perform at least one in vitro functional assay supporting FTY720 dependent electrophisiological changes have impact on the effector functions of Jurkat T-lymphocytes. 

Reply 2. Functional cell assays regarding FTY720 action on lymphocytes have been previously studied. For example, earlier studies have demonstrated that FTY720 can act to work through its binding to sphingosine-1-phosphate receptor subtype 1, which can prevent the egress of lymphocytes from lymph nodes (Morris et al., 2005; Brinkmann, 2009). Importantly, despite the detailed mechanism of FTY720 actions on these KV channels is unclear, if similar findings are experimentally made in lymphocytes occurring in cell culture or in vivo to those presented herein, the additional actions of FTY720 or its structurally related agents described here will result in significant additional changes in immune reactions

Reviewer 2 Report

The ms “Actions of FTY720 (Fingolimod), a Sphingosine-1- 3 Phosphate Receptor Modulator, on Delayed-Rectifier 4 K+ Current and Intermediate-Conductance Ca2+- 5 Activated K+ Channel in Jurkat T-Lymphocytes” presented by Chang et al describes modulation of the immunosuppressive agent 2-Amino-2-(2-[4-octylphenyl]ethyl)-1,3-propanediol (FTY720, fingolimod) of voltage-dependent potassium channel Kv1.3 present in lymphocytes. In relation ith the Immune systeme and the immuno response. 

The experimental setup indeed demonstrates this interaction and the specific modulation of the Kv1.3 channel and the interpretation of the results is straightforward and sound.

The presence of in vivo data to corroborate the in vitro observations would have been perfect to make the story even more significant. 

General comments:

Authors should have included a graphical model/representation of their discussion and not only refer to the graphical abstract.

Section 3.5 can be moved to supplemental material. 

In the experiments where the authors claim that there is no or little effect on the currents in the presence of FTY720. Why don´t they include a measurement with a concentration of FTY720 higher than 3 uM? If the effect is not present and the concentration used is not toxic there should be no change in the obtained result. In this way have a stronger evidence that the presence of FTY720 is minor under the conditions tested.

Why do authors not use of lymphocytes derived from Kv1.3 Null Mice as proper control for all their experiments? This would be very visual especially for the S1P receptor subtype 1 (S1P1) experiments.  

And finally I would like to stress out that the importance of the study on immune response modulators is justified by itself as Asthma, Eczema, Allergic rhinitis, Cancer treatments, organ transplants, Type 1 diabetes, Rheumatoid arthritis, Lupus and AIDS have an important impact on our society on many aspects. It is not necessary to connect it solely to emerging diseases as the virus infection by COVID-19.

I invite authors to remove the COVID-19 statement as it is only mentioned to draw the Journals and push the publication of these results due to the delicate situation we all currently live in. This is not science but politics.

Author Response

Reviewer #2:

The ms “Actions of FTY720 (Fingolimod), a Sphingosine-1- 3 Phosphate Receptor Modulator, on Delayed-Rectifier 4 K+ Current and Intermediate-Conductance Ca2+- 5 Activated K+ Channel in Jurkat T-Lymphocytes” presented by Chang et al describes modulation of the immunosuppressive agent 2-Amino-2-(2-[4-octylphenyl]ethyl)-1,3-propanediol (FTY720, fingolimod) of voltage-dependent potassium channel Kv1.3 present in lymphocytes. In relation ith the Immune systeme and the immuno response. 

The experimental setup indeed demonstrates this interaction and the specific modulation of the Kv1.3 channel and the interpretation of the results is straightforward and sound.

The presence of in vivo data to corroborate the in vitro observations would have been perfect to make the story even more significant. 

General comments:

Question 1. Authors should have included a graphical model/representation of their discussion and not only refer to the graphical abstract.

Reply 1. Thanks for your comment. We added the following sentence on Page 19, The graphic abstract summarized the major findings that through perturbing different types of K+ channels FTY720 may contribute to the functions of immune cells and result in an immune-modulating effect.”

Question 2. Section 3.5 can be moved to supplemental material. 

Reply 2. Thanks for your suggestions. The section of Data analyses has been moved to supplemental material.

Question 3. In the experiments where the authors claim that there is no or little effect on the currents in the presence of FTY720. Why don´t they include a measurement with a concentration of FTY720 higher than 3 uM? If the effect is not present and the concentration used is not toxic there should be no change in the obtained result. In this way have a stronger evidence that the presence of FTY720 is minor under the conditions tested.

Reply 3. In NSC-34 neuronal cells, we found that the amplitude of IK(DR) (i.e., KV3.1-encoded current) was mildly decreased in the presence of FTY720 at a concentration of 3 μM. We did also examine that inhibitory effect of FTY720 (10 μM) on this current in NSC-34 neurons. Current amplitude was noticed to decrease significantly by adding 10μM FTY720. However, the main point is that neither activation nor inactivation time course of the current was altered during the exposure to FTY in NSC-34 neuronal cells.

Question 4. Why do authors not use of lymphocytes derived from Kv1.3 Null Mice as proper control for all their experiments? This would be very visual especially for the S1P receptor subtype 1 (S1P1) experiments.  

Reply 4. Thanks for the specific comments raised by the reviewer. We think that the statement could be interesting. Hence, an additional sentence regarding this issue was included in the revised manuscript. That is, “Whether the presence of FTY720 can perturb ionic currents in lymphocytes derived from KV1.3 null mice remains to be interestingly explored.”

Question 5. And finally I would like to stress out that the importance of the study on immune response modulators is justified by itself as Asthma, Eczema, Allergic rhinitis, Cancer treatments, organ transplants, Type 1 diabetes, Rheumatoid arthritis, Lupus and AIDS have an important impact on our society on many aspects. It is not necessary to connect it solely to emerging diseases as the virus infection by COVID-19. I invite authors to remove the COVID-19 statement as it is only mentioned to draw the Journals and push the publication of these results due to the delicate situation we all currently live in. This is not science but politics.

 Reply 5. Thanks for your comment. We have removed the associated description regarding COVID-19.

Reviewer 3 Report

The manuscript by Chang et al this is a well designed and performed electrophysiological study about the effects of the drug FTY720 on ionic currents in Jurkat T-lymphocytes. In particular, the authors investigated the activity of this molecule on delayed-rectifier K+ current and intermediate-conductance Ca2+-activated K+ channels and the results, if confirmed in vivo, are very promising.

Overall, the manuscript is well written, and I do not find any significant incorrectness. My following comments are of minor character:

Line 39-41: In my opinion, the last sentence of this abstract can be improved. Please, try to rephrase it

Throughout: There are several “spaces”. For instance, in lines 34, 61, 63, 67, 72, 73, 77, 79, 111, 113 etc.  Please check the manuscript.

Line 54: Please add a “space” between “responses” and [5]

Line 55: “Treatment” should be “treatment”

Line 77: The authors describe vey well because they want to study the effect of FTY720 on Kv1.3 channels, but they do not justify the reason to investigate also IKCa. The authors should clarify this in the introduction.

Line 81-83: This sentence is too large and hard to follow. Please rephrase.

Line 206 and 222: These references should be formatted   

Line 330: Maybe “no treated” is best than “without being treated”?

Lines 453-454: “The graphic abstract summarized the major findings”. I am sorry but I can not find any graphic abstract. Where is it?

References: The style of the Reference list needs to be corrected. Please use the ACS reference style

Author Response

Reviewer #3:

The manuscript by Chang et al this is a well designed and performed electrophysiological study about the effects of the drug FTY720 on ionic currents in Jurkat T-lymphocytes. In particular, the authors investigated the activity of this molecule on delayed-rectifier Kcurrent and intermediate-conductance Ca2+-activated K+ channels and the results, if confirmed in vivo, are very promising.

Overall, the manuscript is well written, and I do not find any significant incorrectness. My following comments are of minor character:

Question 1. Line 39-41: In my opinion, the last sentence of this abstract can be improved. Please, try to rephrase it

Reply 1. As per the suggestion by the reviewer, the sentence was rephrased to “The perturbations by FTY720 on different types of K+ channels may contribute to the functional activities of immune cells, if similar findings appear in vivo.”

Question 2. Throughout: There are several “spaces”. For instance, in lines 34, 61, 63, 67, 72, 73, 77, 79, 111, 113 etc.  Please check the manuscript.

Reply 2. The “space” has been corrected in the revised manuscript.

Question 3. Line 54: Please add a “space” between “responses” and [5]

Reply 3. Given that the previous reviewer requested to remove the COVID-19 associated sentence, the indicated sentence has been removed.

Question 4. Line 55: “Treatment” should be “treatment”

Reply 4. Goof! We made a mistake. “treatment” was corrected.

Question 5. Line 77: The authors describe vey well because they want to study the effect of FTY720 on Kv1.3 channels, but they do not justify the reason to investigate also IKCa. The authors should clarify this in the introduction.

Reply 5. As commented by the reviewer, a sentence along with an additional reference was included in the revised manuscript. That is, “Additionally, it was noticed that the activity of intermediated-conductance Ca2+-activated K+ channels was functionally expressed in immune cells [21].”

Question 6. Line 81-83: This sentence is too large and hard to follow. Please rephrase.

Reply 6. The sentence was appropriately rephrased.

Question 7. Line 206 and 222: These references should be formatted   

Reply 7. The references have been formatted.

Question 8. Line 330: Maybe “no treated” is best than “without being treated”?

Reply 8. The sentence was appropriately rephrased to “Distinguishable from the results from control cells (i.e., SEW2871 was not pre-treated), the inactivation time course of IK(DR) remained unaltered in the presence of FTY720.”

Question 9. Lines 453-454: “The graphic abstract summarized the major findings”. I am sorry but I can not find any graphic abstract. Where is it?

Reply 9. The graphic abstract was the original Figure 9. We have corrected it.

Question 10. References: The style of the Reference list needs to be corrected. Please use the ACS reference style

Reply 10. We have revised the format of references according to the journal’s guideline.

Round 2

Reviewer 1 Report

Reviewer has no more concerns.